# Exhaled Aldehydes as Biomarkers for Lung Diseases: A Narrative Review

**DOI:** 10.3390/molecules27165258

**Published:** 2022-08-17

**Authors:** Maximilian Alexander Floss, Tobias Fink, Felix Maurer, Thomas Volk, Sascha Kreuer, Lukas Martin Müller-Wirtz

**Affiliations:** 1CBR—Center of Breath Research, Department of Anaesthesiology, Intensive Care and Pain Therapy, Saarland University Medical Center, 66421 Homburg, Germany; 2Outcomes Research Consortium, Cleveland, OH 44195, USA

**Keywords:** aldehydes, lipid peroxidation, oxidative stress, volatile organic compounds, biomarker

## Abstract

Breath analysis provides great potential as a fast and non-invasive diagnostic tool for several diseases. Straight-chain aliphatic aldehydes were repeatedly detected in the breath of patients suffering from lung diseases using a variety of methods, such as mass spectrometry, ion mobility spectrometry, or electro-chemical sensors. Several studies found increased concentrations of exhaled aldehydes in patients suffering from lung cancer, inflammatory and infectious lung diseases, and mechanical lung injury. This article reviews the origin of exhaled straight-chain aliphatic aldehydes, available detection methods, and studies that found increased aldehyde exhalation in lung diseases.

## 1. Introduction

More than seven million patients die each year from lung diseases, putting enormous socioeconomical burdens on society and health care systems [1]. An early diagnosis may help to initiate treatment at early disease stages to improve treatment outcomes. Identification of biomarkers enabling early diagnosis and treatment is therefore of considerable interest.

Breath analysis could provide rapid, repeatable, and non-invasive diagnosis of numerous diseases by the detection of disease-specific alterations of exhaled volatile organic compounds (VOCs). Although usually patterns of changes in the composition of exhaled air (also referred to as the “exhalome”) help to identify diseased patients, some specific compounds were repeatedly found as potential markers of damage.

A good example is the increased exhalation of straight-chain aliphatic aldehydes in patients suffering from lung diseases. The well-known generation process by lipid peroxidation [2,3,4], high volatility, and good detectability make them interesting candidates as biomarkers to diagnose and monitor progress of lung diseases; especially, since exhaled aldehydes can be measured at the point of care by detection methods such as ion mobility spectrometry and electrochemical sensors [4,5].

Based on the growing importance of exhaled aldehydes in breath research, this article provides a narrative review of the potential use of exhaled aldehydes as biomarkers for lung diseases. Specifically, we review the origin of exhaled straight-chain aliphatic aldehydes, detection methods, and lung diseases previously shown to increase the exhalation of straight-chain aliphatic aldehydes.

## 2. Methods

PubMed and Google Scholar were searched until May 2022 using the following terms: “aldehydes”, “breath”, or “biomarker”, combined with “cancer”, “inflammation”, “infection”, “lung injury”, “COPD”, “asthma”, “COVID”, or “disease”. We included studies that found increased straight-chain aliphatic aldehyde exhalation in lung diseases. Our search was not restricted to specific study designs. In addition, we included relevant literature known to the authors complementing the review article.

## 3. Origin of Straight-Chain Aliphatic Aldehydes

Aldehydes are ubiquitous compounds found in nature and are part of our daily life. They are highly reactive and consist of a carbonyl group attached to at least one hydrogen atom. They are represented as R-CHO, where R is an attached group, either aromatic or aliphatic. This review focusses on C2 to C10 straight-chain aliphatic aldehydes, as listed in Table 1.

Major sources for exogenous exposure to aldehydes are biomass and fossil fuel combustion, vehicle exhaust, power plants and wood burning fumes. For example, acetaldehyde and formaldehyde can be frequently detected in the surrounding air. However, smoking and the intake of alcohol are also major causes for exposure to aldehydes [6].

A major endogenous source for aldehyde generation is lipid peroxidation which is triggered by oxidative stress [7]. In healthy individuals a balance between oxidative and antioxidative mechanisms exists. When this balance is disturbed by diseases or injuries, reactive oxygen species and free radicals cause damage. Radicals oxidize and degrade polyunsaturated fatty acids of lipid membranes—a process called lipid peroxidation (Figure 1) [7,8].

Straight-chain aliphatic aldehydes are some of the most abundant products of lipid peroxidation [2,3,4,9] and are exhaled in the lower parts-per-billion (ppb) concentration range [10,11]. Exhaled aldehydes were thus repeatedly investigated as volatile biomarkers of lipid peroxidation-inducing diseases (see Section 5, Section 6, Section 7 and Section 8).

**Figure 1 molecules-27-05258-f001:**
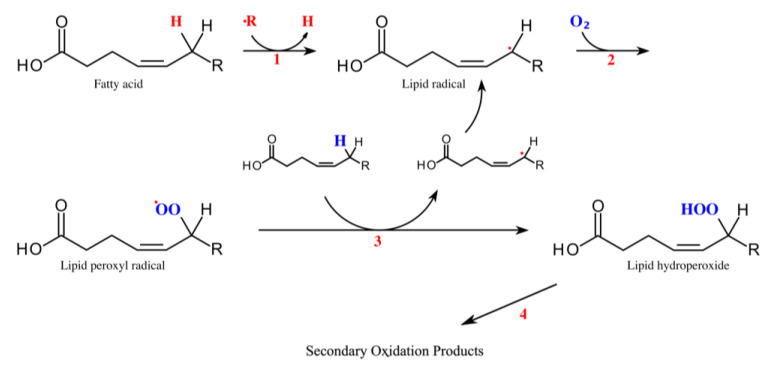
Lipid peroxidation. Modified from [12]. Two steps are essential for lipid peroxidation—initiation (1) and propagation (2). During initiation, a hydrogen atom is removed from the fatty acid forming a lipid radical (1). This can happen through enzymatic reactions from lipoxygenases, hydroperoxide-lyases and peroxygenases or by non-enzymatic processes. During propagation, the lipid radical reacts with oxygen which produces a peroxyl radical (2). In the next step the peroxyl radical reacts with another unsaturated lipid. It abstracts a hydrogen atom to form a hydroperoxide radical and a new lipid radical (3). Hydroperoxide radicals are unstable and quickly react to form other radicals and secondary products. In further cyclization reactions and cleavages, different compounds are produced including straight-chain aliphatic aldehydes (4) [8].

## 4. Detection Methods for Exhaled Aldehydes

Several methods were used to detect straight-chain aliphatic aldehydes in breath. We will present a comprehensible overview on previously used methods in the following. Readers that are interested in an in-depth review on detection methods for volatile organic compounds are referred to the excellent methodological review by Buszewski et al. [13].

In general, gas chromatography–mass spectrometry (GC-MS) is considered the gold standard for the measurement of volatile organic compounds in breath. Large GC-MS databases enable the exact identification of an analyte according to retention time and molecular mass [13,14]. Given the universal applicability, several studies used GC-MS to detect aldehydes in breath. Further mass spectrometry methods, such as selected ion flow tube–mass spectrometry (SIFT-MS) or time of flight–mass spectrometry (TOF-MS) allow rapid real time measurements of exhaled aldehydes [13,15,16].

A downside of mass spectrometry systems is the bulky and expensive setup making point-of-care applications infeasible. Therefore, more portable systems were previously used for aldehyde detection in breath. For example, multi-capillary coupled–ion mobility spectrometry (MCC-IMS) can be applied at point-of-care [13,17], and has been used to detect aldehydes in exhaled breath in experimental and clinical settings [4,18,19].

Due to recent findings identifying aldehydes as potential breath biomarkers, electrochemical sensors were developed to further simplify point-of-care application. Obermeier et al. developed a combined sensor for aldehydes, carbon monoxide and nitric oxide and showed feasibility of continuous aldehyde monitoring in pigs [5]. In addition, the sensor enabled the identification of patients suffering from diabetes or lung cancer in a first pilot study [5]. Most recently, a zinc oxide nanowire sensor was developed for the detection of aldehydes down to 0.6 ppm which still needs optimization [20], as exhaled aldehydes are usually exhaled in the lower ppb concentration range [11]. Apart from good applicability for point-of-care analysis, specificity of these sensors for aldehydes remains unclear. As breath contains about 1500 volatile organic compounds, cross reactions of electrochemical sensors are likely [21]. There is thus a further need to develop and optimize point-of-care methods for aldehyde sensing.

## 5. Aldehyde Exhalation and Lung Cancer

Cancer is a leading cause of death with lung, breast and colorectal cancer contributing the most [22]. Early detection is essential to improve survival by initiation of treatments at early stages.

Screening methods for lung cancer have been debated for many years. Several biomarkers in blood and sputum have been investigated including DNA, RNA, circulating tumor cells, proteins and autoantibodies [23]. Currently, there is no established molecular biomarker used in clinical practice for early detection of lung cancer. Known markers, such as NSE, CEA and CA125 have poor sensitivity and specificity [24]. Combinations of biomarkers might improve the diagnostic value but larger multi-centric validation studies are pending [25,26].

Chest x-ray as a screening method showed no reduction in lung cancer deaths [27,28]. In addition, sensitivity of about 80% was reported in a recent review with almost 20% of lung cancer patients not being detected by chest x-rays [29].

Crucial for today’s screening was a study conducted by the National Lung Screening Trial Research Team. 53,454 patients were enrolled in a large multi-center study comparing the diagnostic value of a conventional chest x-ray versus a low dose computed tomography (LDCT). LDCT resulted in a reduction of mortality of up to 20% in high-risk patients [30]. The sensitivity for detecting lung cancer by LDCT is greater than 80% for which reason LDCT is the primary screening method for lung cancer in various countries despite risks of radiation exposure and overdiagnosis [31].

In contrast to the above-mentioned diagnostic methods, an optimal screening tool should be fast, cost-effective, and preferably non-invasive. All these requirements are provided by breath analysis.

“Cancer is a large group of diseases that can start in almost any organ or tissue of the body when abnormal cells grow uncontrollably, go beyond their usual boundaries to invade adjoining parts of the body and/or spread to other organs“ is the definition for cancer by the World Health Organization [32]. The “uncontrollably” “invasive” growth is accompanied by high metabolic activity. The increased metabolic activity resulting from the uncontrolled and fast growth leads to an increased production of reactive oxygen species, for instance, during oxidative phosphorylation in mitochondria, increased enzymatic activity and modified metabolism in cancer cells [33,34]. Consistently, increased concentrations of aldehydes as products of oxidative stress were detected in the headspace of cultured lung cancer cells [3], and increased aldehyde exhalation was repeatedly reported in lung cancer patients (Table 2).

As one of the first in 2010, Fuchs et al. measured the concentrations of aldehydes in 12 lung cancer patients, 12 healthy smokers and 12 non-smoking healthy subjects. Exhaled concentrations of hexanal, pentanal, octanal and nonanal were significantly higher in lung cancer patients than in smoking or non-smoking healthy individuals. Propanal, butanal, heptanal and decanal concentrations did not differ between the groups [35].

In the same year, Poli et al. published results from measurements of aldehydes in exhaled breath of 40 lung cancer patients compared to 38 healthy individuals. They measured the exhaled concentrations of C3-C9 aldehydes. All measured aldehydes were significantly higher in the ex- and non-smoking lung cancer patients compared to controls, except for propanal, which was characteristic for smoking lung cancer patients. For example, the median exhaled concentration of butanal in lung cancer patients was more than twice as high as in controls (10.8 pM vs. 26.2 pM) and hexanal concentration was three times greater in lung cancer patients compared to controls (10.3 pM vs. 38.1 pM). By using this set of aldehydes, 90% of lung cancer patients and 92% of controls were classified correctly [36].

Buszewski et al. measured exhaled concentrations of aldehydes in 44 healthy individuals and 29 lung cancer patients. Propanal was increased in lung cancer patients and again in smokers. Butanal was significantly increased in lung cancer patients compared to healthy individuals and smokers. Other aldehydes were not measured [37].

Handa et al. analyzed 115 different volatile organic compounds measured in 50 patients with lung cancer and 39 healthy controls. Ten peaks were significantly higher in lung cancer patients including hexanal, heptanal and nonanal. Nonanal additionally allowed the differentiation between squamous cell carcinoma and adenocarcinoma [38]. Consistently, Baumbach et al. found increased concentrations of nonanal obtained during bronchoscopy in lung cancer patients [39].

Corradi et al. and Ulanowska et al. conducted two of the largest studies that found increased aldehyde exhalation in 138 and 137 lung cancer patients, respectively. Corradi et al. measured increased concentrations of heptanal in lung cancer patients. Interestingly, exhaled concentrations of other aldehydes were not increased [40]. One reason for this might be the composition of the control group which included patients with lung diseases other than cancer. As outlined in the following chapters, inflammatory and infectious lung diseases influence aldehyde exhalation which might have diminished the difference of aldehyde exhalation in comparison to lung cancer patients. In contrast, Ulanowska et al. found increased exhaled concentrations of propanal in lung cancer patients compared to healthy individuals. Furthermore, pentanal and hexanal were only detectable in cancer patients but not in healthy individuals [41].

Finally, a recently published investigation included 157 lung cancer patients and 368 healthy individuals. Pentanal, hexanal, heptanal, octanal, nonanal and decanal exhalation was increased in lung cancer patients. Important in this study is the perioperative setting. Breath sampling was performed immediately before surgery to minimize impact of external factors such as environmental contaminations or prior food intake [42].

In contrast to the above presented findings, some studies could not show increased aldehyde exhalation in lung cancer patients. For example, Callol-Sanchez et al. screened 81 lung cancer patients and 83 healthy control patients explicitly for exhaled aldehydes but did not detect a difference [43].

Several reasons might explain the different findings throughout the literature. First is the subject-specific influence on aldehyde exhalation, as sex-related differences in propanal exhalation were reported [44]. Thus, results might be biased by unbalanced baseline characteristics of the assessed study populations. For example, most healthy subjects included in the study of Fuchs et al. were between 20 and 30 years old, whereas all lung cancer patients were older than 50 years [35]. Larger studies on exhaled aldehydes as biomarkers for lung cancer with more than 100 subjects, providing more balanced baseline characteristics, are still rare [40,41,42]. Future studies may thus focus on influences of baseline characteristics such as age, sex, or comorbidities on aldehyde exhalation.

Second, different sampling methods were used throughout the presented studies. Tedlar bags were mostly used but also Bio-Voc systems were used. Probands exhale into these systems and the breath sample is transferred to the respective analysis device. Although made from inert materials, it was shown that used Tedlar bags may release volatile organic compounds from previous usage which highlights the interactions between sampling material and analytes [45]. Furthermore, Tedlar bag samples usually contain mixed exhaled air as opposed to Bio-Voc samples which mostly contain alveolar air. Thus a previous study found 137 VOCs with Tedlar bags compared to only 47 VOCs with the Bio-Voc system [46].

Third, contaminations from the surrounding, diet and medication may considerably alter the results from breath analysis. For example, Kischkel et al. found significantly increased exhaled aldehyde concentrations in lung cancer patients, which after correction for inspired concentrations was no longer significant [44]. Furthermore, previous exposition to fumes or disinfectants present in the hospital environment alters the composition of breath aldehydes, as for example propanal is a typical ingredient of disinfectants [47,48]. To minimize potential influences from diet and environmental contamination, sampling in the perioperative setting could thus be favorable [42].

Finally, different statistical methods and algorithms might produce different statistical significances. For example, Poli et al. performed in addition to analysis of variance between groups a multivariate analysis using a structure matrix and cross-validation and implementing factor scores for establishing a predictive algorithm [36]. Ulanowska et al. used discriminant analysis and a CHAID tree to test for statistical significance [41], whereas Handa et al., for example, didn’t use predictive models at all [38].

In summary, several studies found increased aldehyde exhalation in lung cancer patients. For assessing the usefulness of aldehydes as biomarkers for lung cancer, future studies should have standardized breath sampling methods, use rather large study populations, and eliminate contaminants from the surroundings. Finally, it remains unclear whether breath analysis is useful to screen for early disease stages, as current studies only included patients with lung cancer already diagnosed with established diagnostics potentially missing early stages.

**Table 2 molecules-27-05258-t002:** Aldehydes as biomarkers of lung cancer.

Author/Study	Method	Cancer Stage *	Histologic Type	Substance	Patients/ Controls [*n*]	Concentration Ratio Sick/Healthy
Fuchs et al. (2010) [35] Breath gas aldehydes as biomarkers of lung cancer	GC-MS	>T3	NSCLC	Pentanal Hexanal Octanal Nonanal	12/24	9.5 - 4.7 7.2
Poli et al. (2010) [36] Determination of aldehydes in exhaled breath of patients with lung cancer by means of on-fiber-derivatisation SPME–GC/MS	GC-MS	Stage 1 & 2	NSCLC	Butanal Pentanal Hexanal Heptanal Octanal Nonanal	40/38	2.4 2.2 3.7 2.3 2.0 3.6
Baumbach et al. (2011) [39] Significant different volatile biomarker during bronchoscopic ion mobility spectrometry investigation of patients suffering lung carcinoma	IMS	-	-	Nonanal	19	-
Ulanowska et al. (2011) [41] The application of statistical methods using VOCs to identify patients with lung cancer	GC-MS	-	SCLC, NSCLC, Others	Propanal Pentanal Hexanal	137/143	1.1 5.9 4.5
Buszewski et al. (2012) [37] Identification of volatile lung cancer markers by gas chromatography–mass spectrometry: comparison with discrimination by canines	GC-MS	-	SCLC, NSCLC	Butanal	29/44	-
Handa et al. (2014) [38] Exhaled Breath Analysis for Lung Cancer Detection Using Ion Mobility Spectrometry	IMS	≥Stage 1	NSCLC	Hexanal	50/39	-
				Heptanal Nonanal		
Corradi et al. (2015) [40] Exhaled breath analysis in suspected cases of non-small-cell lung cancer: a cross-sectional study	GC-MS	≥Stage 1	NSCLC	Heptanal	71/67	1.3
Schallschmidt et al. (2016) [49] Comparison of volatile organic compounds from lung cancer patients and healthy controls—challenges and limitations of an observational study	GC-MS	≥Stage 1	-	Propanal Butanal Pentanal Hexanal Decanal	37/23	3.3 2.0 1.5 1.1 2.7
Wang et al. (2022) [42] Identification of lung cancer breath biomarkers based on perioperative breathomics testing: A prospective observational study	TOF-MS	-	SCLC, NSCLC	Pentanal Hexanal Heptanal Octanal Nonanal Decanal	157/368	-

Concentration ratios are missing for reports that did not provide mean or median concentrations. * Classification according to American Joint Committee on Cancer (AJCC) TNM system. Abbreviations: GC-MS, Gas Chromatography–Mass Spectrometry. IMS, Ion Mobility Spectrometry. TOF-MS, Time-Of-Flight Mass Spectrometry. NSCLC, Non-Small Cell Lung Cancer. SCLC, Small Cell Lung Cancer.

## 6. Aldehyde Exhalation and Inflammatory/Infectious Lung Diseases

Oxidative stress is fundamental to inflammation [50]. During inflammatory responses, neutrophils produce large amounts of reactive oxygen species to counteract infection [51,52]. Reactive oxygen species and resulting tissue damage cause lipid peroxidation. Consistently, lipid peroxidation products in breath correlate with cytokines in bronchoalveolar lavage fluid [53]. Thus, exhaled aldehydes as well-known products of lipid peroxidation were targeted as biomarkers for diagnosing patients suffering from inflammatory lung diseases (Table 3).

Chronic obstructive pulmonary disease (COPD) is a disease affecting millions of people worldwide. Current diagnosis and therapy monitoring is mainly based on spirometry [54]. Main disadvantages of this method are decreased sensitivity and lack of monitoring in early disease stages [55]. Exhaled biomarkers might have a better sensitivity especially in early disease stages and could complement the monitoring of disease progress [56].

For example, Corradi et al. found increased aldehyde exhalation in patients suffering from chronic obstructive pulmonary disease (COPD) [57]. Compared with 20 non-smoking healthy individuals, exhaled hexanal and heptanal concentrations were increased in 20 COPD patients and in 12 smokers with no differences between COPD patients and smokers [57]. In contrast, exhaled concentrations of hexanal and heptanal did not differ between 10 healthy and 12 asthmatic children [58]. Interestingly, nonanal exhalation was decreased in children suffering from asthma, while being unaffected by COPD.

Due to the ongoing COVID-19 pandemic much effort was spent on how to diagnose SARS-CoV-2 infections efficiently. The diagnostic gold standard for COVID-19 is multiplication and detection of viral RNA by polymerase chain reaction (PCR), which shows high specificity and sensitivity [59]. Despite its advantages, PCR is time consuming, expensive, needs specialized laboratories, and might provide false-negative results due to sampling errors from nasopharyngeal swap sampling [60]. Antigen tests provide an inexpensive, portable and fast method for diagnosing COVID-19 but studies showed only a limited sensitivity especially in asymptomatic patients [61,62]. For example, Jegerlehner et al. reported a sensitivity of 40% for detecting SARS-CoV-2 infections in asymptomatic patients with antigen tests [61].

Infections cause inflammation. Infectious lung diseases may thus similarly alter the exhalome as do inflammatory lung diseases. Breath analysis could therefore provide a fast, non-invasive, and potentially more sensitive diagnostic than established methods at point of care. Interestingly, three recent pilot studies investigated the exhalome for markers of an infection with SARS-CoV-2 and reported increased aldehyde exhalation.

Ruszkiewicz et al. performed two independent observational prevalence studies in two cities—Dortmund (Germany) and Edinburgh (United Kingdom) [60]. They measured concentrations of ethanal, octanal and heptanal by means of GC-MS. 98 patients were recruited of whom 10/65 (Dortmund) and 21/33 (Edinburgh) were positive for SARS-CoV-2. Under inclusion of other VOCs, the authors were able to identify SARS-CoV-2 infections by computing stratification models with an area under the receiver operating characteristic curve (AUROC) of 0.91 (95% confidence interval (95%CI): 0.87 to 1) (Dortmund) and 0.87 (95%CI: 0.67 to 1) (Edinburgh).

Berna et al. aimed to diagnose COVID-19 in children by means of breath analysis [63]. 84 VOCs were detected in the exhaled breath of 26 children of whom 11 had tested positive for SARS-CoV-2 infections. Consistent with the above presented results of Ruszkiewicz et al., heptanal and octanal were significantly increased. In addition, the exhalation of nonanal was increased.

Grassin-Delyle et al. analyzed the breath of 40 mechanically ventilated patients suffering from acute respiratory distress syndrome (ARDS) of whom 28 had COVID-19 [64]. By use of machine learning algorithms, they determined a specific breath print for COVID-19, which included nonanal among in total four VOCs. Discrimination between COVID ARDS and non-COVID ARDS was possible with an accuracy of 93% (sensitivity: 90%, specificity: 94%, AUROC: 0.94–0.98).

The above-described initial observations suggest that pulmonary inflammation affects aldehyde exhalation. However, currently available studies are small, and it remains unclear how the severity of the disease and initiated treatments influence aldehyde exhalation. Since aldehyde exhalation is affected by smoking and non-pulmonary diseases (see Section 8), it seems particularly interesting to monitor the course of aldehyde exhalation during disease progress and ongoing treatments. Alterations in aldehyde exhalation, most probably a decrease throughout treatment, could help to evaluate treatment success or disease progress.

## 7. Aldehyde Exhalation and Mechanical Lung Injury

Mechanical ventilation injures the lung by exposing alveolar tissue to increased stress and strain causing cell membrane damage, ruptures of intercellular contacts and destruction of the extracellular matrix [65,66,67]. When mechanical injury outweighs cellular repair mechanisms, cell apoptosis or necrosis follows [67]. Initial mechanical injury is then further aggravated by the ensuing inflammatory response [68,69]. High inspired oxygen concentrations, often used for mechanical ventilation to counteract impaired gas exchange additionally increase the production of reactive oxygen species causing further damage [70,71].

Early detection of harmful ventilation could help clinicians to improve ventilatory settings even before severe lung injury occurs. Systemic and local inflammatory markers or histologic signs of lung injury were previously used to detect harmful ventilation but are invasive and pathological results appear late in the injury cascade [72,73,74]. Thus, there is currently no reliable non-invasive method for early detection of harmful ventilation.

Breath analysis has the potential to fill this gap. A recent series of studies identified the exhaled aldehyde pentanal as a potential biomarker for ventilator-induced lung injury in rats [18] (Table 3) which is consistent with the previous finding that harmful ventilation increases lipid peroxidation in lung tissue [69]. It might be expected that high inspired oxygen concentrations used during mechanical ventilation may additionally trigger pentanal exhalation but exhaled concentrations of pentanal were unaffected over a wide range of inspired oxygen concentrations during 12 h of mechanical ventilation in rats [75]. Finally, a pilot study showed the feasibility of exhaled pentanal monitoring in mechanically ventilated patients [4].

Most interestingly, exploratory analyses revealed a significant association of mechanical power—a clinical measure for the invasiveness of mechanical ventilation—and pentanal exhalation in rats as well as in human subjects [4,18]. Future clinical studies should evaluate whether exhaled pentanal is a useful biomarker to monitor the invasiveness of mechanical ventilation.

**Table 3 molecules-27-05258-t003:** Aldehydes as biomarkers of inflammatory/infectious lung diseases and mechanical lung injury.

Author/Study	Method	Substance	Patients/ Controls [*n*]	Concentration Ratio Sick/Healthy
Corradi et al. (2003) [57] Aldehydes in exhaled breath condensate of patients with chronic obstructive pulmonary disease	LC-MS	Hexanal Heptanal	20/32	3.6 1.9
Ruszkiewicz et al. (2019) [60] Diagnosis of COVID-19 by analysis of breath with gas chromatography- ion mobility spectrometry—a feasibility study	SIFT-MS	Ethanal Heptanal Octanal	27/63	-
Berna et al. (2021) [63] Reproducible breath metabolite changes in children with SARS-CoV-2 infection	TOF-MS	Heptanal Octanal Nonanal	15/10 12/12	-
Grassin-Delyle et al. (2021) [64] Metabolomics of exhaled breath in critically ill COVID-19 patients: A pilot study	TOF-MS	Nonanal	28/12	-
Müller-Wirtz et al. (2021) [18] Volutrauma Increases Exhaled Pentanal in Rats: A Potential Breath Biomarker for Ventilator-Induced Lung Injury	MCC-IMS	Pentanal	150 ^#^	- *
Müller-Wirtz et al. (2021) [75] Differential Response of Pentanal and Hexanal Exhalation to Supplemental Oxygen and Mechanical Ventilation in Rats	MCC-IMS	Pentanal Hexanal	30 ^#^	- *
Müller-Wirtz et al. (2021) [4] Quantification of Volatile Aldehydes Deriving from In Vitro Lipid Peroxidation in the Breath of Ventilated Patients	MCC-IMS	Pentanal	12	-

Concentration ratios are missing for reports that did not provide mean or median concentrations. * Observation of increasing aldehyde exhalation over time under continuous monitoring. # Total number of analyzed animals. Abbreviations: LC-MS, Liquid chromatography–mass spectrometry. SIFT-MS, Selected ion flow tube mass spectrometry. TOF-MS, Time-of-flight mass spectrometry. MCC-IMS, Multi-Capillary Column—Ion Mobility Spectrometry.

## 8. Aldehyde Exhalation from Non-Pulmonary Diseases

Oxidative stress—the most important trigger for lipid peroxidation—is part of the pathogenesis of most diseases [50,76,77]. Therefore, non-pulmonary diseases can contribute to aldehyde exhalation, as shown for extrapulmonary cancer including gastric, colorectal and breast cancer [78,79,80,81,82,83]. Other diseases, such as diabetes and chronic kidney failure, also increase aldehyde exhalation [84,85]. Many other diseases such as cardiovascular diseases or local inflammation are related to increased production of reactive oxygen species, and could contribute to aldehyde exhalation [86,87].

Exhaled aldehydes are thus non-specific markers for various diseases. Although it is likely from an anatomical perspective that pulmonary as opposed to non-pulmonary diseases are more dominant sources of aldehyde exhalation, future studies should further clarify the relative contributions of pulmonary and non-pulmonary sources to aldehyde exhalation. For a valid comparison of exhaled aldehyde concentrations between pulmonary and non-pulmonary diseases, breath sampling and analysis would have to be carried out with the same standardized methods. Due to a variety of methods and settings used for breath analysis, it is currently impossible to compare exhaled aldehyde concentrations across studies.

## 9. Conclusions

The measurement of straight-chain aliphatic aldehydes in breath provides a fast and non-invasive diagnostic method for the detection and monitoring of various pathologic conditions of the lung. Ion mobility spectrometry or newly developed electrochemical aldehyde sensors are applicable at point-of-care, considerably lowering technical burdens. However, breath sampling methods strongly differ which limits the comparability of studies in breath research. International technical standards are thus highly needed.

Local pathologies of the lung have most likely the strongest impact on aldehyde exhalation, while extrapulmonary diseases also contribute. Measurement of exhaled aldehydes could be useful to detect and monitor lung cancer and inflammatory and infectious lung diseases. Recent experimental studies indicate a potential use to monitor the invasiveness of mechanical ventilation.

Of note, the unspecific nature of aldehyde generation makes the measurement of exhaled aldehydes at a single time point less meaningful. More promising is the monitoring and interpretation of aldehyde exhalation over time and the interpretation of exhaled aldehyde concentrations in the light of the overall clinical picture. Exhaled aldehydes could then help to evaluate disease progress and treatment success in patients suffering from lung diseases.

## Figures and Tables

**Table 1 molecules-27-05258-t001:** Overview of straight-chain aliphatic aldehydes.

Aldehyde	Chain Length	Structural Formula
Ethanal	C2	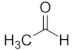
Propanal	C3	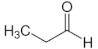
Butanal	C4	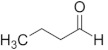
Pentanal	C5	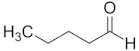
Hexanal	C6	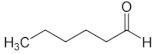
Heptanal	C7	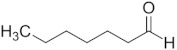
Octanal	C8	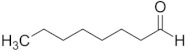
Nonanal	C9	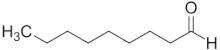
Decanal	C10	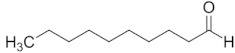

## Data Availability

Not applicable.

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
