# Peer review of "Exhaled Aldehydes as Biomarkers for Lung Diseases: A Narrative Review"

_molecules, 2022, doi:10.3390/molecules27165258_

Round 1

Reviewer 1 Report

Description of aldehydes for the detection of different lung diseases is well organized by disease, which makes it clear to focus on the specific condition. 

To maximize the comprehensiveness of the review it is advised to briefly describe the current diagnostic availability for each disease and limitations, and how a breath test would benefit the diagnostic pathway. This has been highlighted only in point "7, Aldehyde Exhalation and Mechanical Lung Injury".

Please provide a rationale on the focus of Aldehydes from C2 to C10.  

Reviewer 2 Report

I suggest to clarify in the methods the following points

what exactly the research focused on,

what kind of studies were being sought, whether randomized ones or not

and what were the main objectives detected by them.

I suggest to include the following referenes to widen the discussion about exhaled volatile compounds, mass spectrometry and cigarette smoking:

- Breath Res. 2018 Aug 6;12(4):046007. 

-Biochim Biophys Acta Mol Basis Dis. 2021 Jan 1;1867(1):165990. 

Reviewer 3 Report

This article is a review of breath aldehydes as biomarkers in lung disease.

It is written very carefully, and it is thought that much of the data is covered.

Discussed in three groups: lung cancer, respiratory infections, and mechanical lung cancer, each unit is well marked. In addition, the section on lung cancer, in particular, has a lot of data and is thoroughly discussed, including issues. There is also a mention of non-pulmonary diseases.

By the way, is it possible to compare the disease groups in terms of the absolute amount of aldehydes in breath for these three groups?

The conclusion is "to detect and monitor lung cancer and inflammatory and infections lung diseases", but is it really possible to distinguish between each disease group as detection?

As a specific example, for example, will it be replaced by chest X-rays in health checkups? "It would be great if you could touch on a few examples."

Reviewer 4 Report

1. In the Methods section, you must indicate the period for which the articles were selected, clarify the inclusion / non-inclusion criteria in the study.

2. Table 1 contains known information, there is no need to include it in the article.

3. Section 4 needs to be specified, preferably in the form of a table that summarizes the methods of determination and their analytical characteristics, the list of determined VOCs, etc.

4. Regarding lung cancer, I would also like to see a table that summarizes the studies, the number of patients, diagnostic characteristics, limitations, including for which histological types/stages, etc. the results are applicable. In the form in which the data are presented, they are not convenient for analysis.

5. I would like to see an analysis and comparison with other diagnostic methods, a description of the advantages / disadvantages, this will improve the quality of the review.

Round 2

Reviewer 4 Report

I have no more comments on the article. I think that in its present form the article can be recommended to be accepted for publication.